# Purposeful Outdoor Learning Empowers Children to Deal with School Transitions

**DOI:** 10.3390/sports7060134

**Published:** 2019-05-31

**Authors:** Vikki Slee, John F. Allan

**Affiliations:** Carnegie School of Sport, Leeds Beckett University, Leeds LS6 3QS, UK

**Keywords:** school children, transitions, primary and secondary school, nature, tailored outdoor education programming, individuality, adaptable productive functioning, green spaces, health and psychological well-being, self-determination

## Abstract

UK schoolchildren are vulnerable to transitional stress between primary and secondary school, which may impact negatively upon their psychological health and academic achievement. This is experienced most acutely by children from ethnic minorities and lower socio-economic status (SES) households. Outdoor Adventure (OA) residential programmes are purported to develop behavioural adaptations which enable positive educational transitions of children. Personal, social and academic skills (self-reliance, getting along with others, curriculum alignment) may be best acquired through bespoke nature-based residential OA programmes. A mixed methods study evaluated the efficacy of a bespoke OA programme for developing school children’s psychological well-being and self-determination during their transition into secondary school. Participants were representantives of ethnic minorities and lower SES groups. A bespoke OA residential programme achieved the strongest scale of change in children’s psychological well-being (F (30,69) = 1.97 < 0.05) and self-determination (effect size 0.25) compared to a generic OA residential and a non-OA school-based induction programme. Qualitative testimonies illuminated personal experiences and processes underpinning the perceived changes in the self-determination domains of *Autonomy* (the capacity to self-direct learning), *Competence* (the ability to complete tasks) and *Relatedness* (developing connections with others). Providing early opportunities for children to take control for their own learning through nature-based tailored OA programming improves their psychological well-being and adaptability to combat transitional stress.

## 1. Introduction

### 1.1. Transitions into Secondary School

The transition between primary and secondary school represents a significant adjustment for young people both socially and academically. Periods of transition typically necessitate life change which may cause vulnerability to psychological adjustment difficulties [1,2]. Challenges include moving from a small school where lessons take place in a single room with one teacher, to navigating a complex timetable with multiple subjects, locations and teachers. Although many children make successful transitions, this process can impact a significant minority of individuals to such an extent that their level of motivation, psycho-social well-being and subsequent academic attainment is impaired [3,4,5,6,7,8,9,10]. Moreover, the potential harm initiated by school-based educational transitions may be experienced most acutely by children already considered vulnerable to life changes—namely, those from lower socio-economic status (SES) households and ethnic minority groups [11,12]. 

Given the extent of challenges presented during educational transitions, several studies have aimed to understand factors associated with transitional stress and the successful assimilation of school children (e.g., [1,13]). Risk factors associated with transition vulnerability include the loss of old friendships, feelings of isolation, and lack of curriculum continuation. Schools that report successful transitions help children to develop new friendships, become more self-reliant, and promote curriculum interest and continuity [14,15]. These practical findings fit well with the strongest contemporary theories about the adaptive capabilities of young people in overcoming problems [16,17,18]. For example, the Self-Determination Theory (SDT) proposes that conditions which support high levels of autonomy (feelings of independence and control), competence (a sense of accomplishment) and relatedness (ability to get along with others) will help to nurture high-quality forms of motivation, which underpins wellness in that setting [19,20,21]. 

To further develop an understanding of school children’s prerequisites for successful educational transitions, there have been calls for longitudinal research which reveals the processes which underpin their behavioural adjustment. Longitudinal studies offer opportunities to investigate risk and protective factors that co-occur and affect transition success over time [8]. Such developed insight may help to inform practices which provide young people with a broad set of assets and resources as a focus for positive change alongside skills built for risk avoidance or amelioration. 

### 1.2. Outdoor Adventure (OA) Programming

There has been extensive research implicating OA residential programming for promoting positive behavioural adaptations pertinent for educational transitions. These include short-term and lasting improvements in self-efficacy, social connectedness, problem solving, resilience and academic performance [22,23,24,25,26,27]. It is claimed outdoor learning generates ‘social capital’ by boosting self-confidence and creativity [28], fostering pride and a sense of belonging [29,30], and improving cooperation, honesty, trust and compassion [31,32]. 

OA relies upon the process of experiential learning within a dynamic, natural setting to generate adaptive skill sets for young people. Experiential learning represents a progressively staged mechanism whereby participants learn through direct immersion and reflection of experiences [33,34]. An array of unfamiliar experiences is purported to compel individuals to engage with risk and uncertainty [35]. This helps to create an authentic sense of capability initiated through a supported, controlled disruption of everyday behaviours [21,36]. 

Additionally, a growing number of studies suggest just being in close proximity to green spaces significantly contributes to improvements in children’s mental health and wellbeing [37,38,39,40]. Indeed, even short-term doses of nature can make a marked impact upon their mental health in natural settings. Just five minutes of exercise undertaken in an urban green space may be sufficient to boost physical and mental well-being of young people through ‘biophilia’ [41], described as an innate connection to nature [42,43]. This has particular significance given a growing number of children, particularly from poorer households and ethnic minority groups, have less opportunities to experience natural environments [44,45]. Further, the most recent meta-analysis advocating the health-giving benefits of greenspace exposure recommend policymakers and practitioners create, maintain and improve greenspaces specifically in deprived areas [46]. 

Because of the perceived benefits associated with OA and young people, schools have deployed a range of OA providers to deliver residential programmes which aim to boost children’s capacity to avoid any transition-related problems [47,48,49]. It is proffered personal and social skills which contribute to the effective transition of school children may be best optimised through carefully constructed residential experiences, incorporating challenges which are shaped to accommodate the specific abilities of participants. This includes purposeful exposure of children to nature-based OA, whereby teachers collaborate with experienced OA providers to shape programming which meets the aims of schools [50]. 

Irrespective of the apparent success of different forms of OA programming, there is little empirical research to show how OA empowers schoolchildren to cope with educational transitions. It is argued beneficial outcomes emanating from OA may be largely based on intuitive belief systems, situation-specific novelty or even coercion, rather than an informed understanding of the dynamic nature of processes and outcomes. Therefore, any benefits will remain context-specific [51] and may not readily transfer into everyday behaviours in schools [52,53,54]. Without evidence of the transitional experiences of schoolchildren, coupled with the processes within OA programming which influence changes in young people (i.e., facilitation techniques, group dynamics), OA practitioners and schools are unlikely to make judgements about the validity of OA programmes to meet school aims. Addressing this shortcoming may ensure that provision meets the needs of particular groups, such as pupils newly transiting into secondary school. 

### 1.3. Research Aim and Objectives 

The aim of this study was to investigate the efficacy of three contrasting induction programmes for facilitating improvements in children’s psychological well-being and self-determination during their transition into secondary school. 

Objectives:(i)Evaluate the psychological well-being of schoolchildren pre- and post-induction programmes in three conditions: Tailored OA, School-based non-OA and Generic OA programmes.(ii)Evaluate the self-determination of schoolchildren pre- and post-induction programmes in three conditions Tailored OA, School-based non-OA and Generic OA programmes.(iii)Investigate the processes associated with schoolchildren’s learning within a tailored OA programme.(iv)Evaluate the sustainability of the tailored OA programme four months later.

## 2. Methods 

### 2.1. Study Overview

Partnerships between educational providers and outdoor practitioners are deemed critical for educating children through OA [55]. Therefore, this study was a collaborative project between three UK secondary schools, a university and an Adventure Development Unit within a Metropolitan District Council.

### 2.2. Participants 

Following institutional ethical approval, 100 school children, mean (M) age of 11 years (female N = 55/55%), were recruited as a purposive sample of pupils transiting into three inner-city secondary schools in the North of UK. The majority of pupils were representatives of lower SES households (classes 5 and 6) and ethnic minorities.

The sample comprised: (i)A main ‘Tailored OA’ group of 60 pupils who attended an OA residential programme specifically designed to promote skills recognised as important for children’s school-based transition, (female N = 32/53%).(ii)A ‘School Induction’ group of 20 pupils who experienced active, classroom-based activities designed to integrate them into the school system (female N = 11/55%).(iii)A ‘Generic OA’ group of 20 pupils who attended a pre-existing generic OA residential (female N = 12/60%). Across all groups, participants were from (SES households and of diverse ethnicity.

Three school teachers, mean (M) age of 26.4 (± SD1.56) of white ethnicity, (female, N = 2, 67%) were sampled to ascertain their perceptions associated with pupils’ immersion within the ‘Tailored OA’ group. 

### 2.3. Procedures and Programme Design

#### 2.3.1. Tailored OA Programme 

The Tailored OA group of transiting children participated in a three-day, two-night residential in a local authority OA centre. Emphasis was continually placed on the school’s ideology to develop self-determined learners and promote the core values of honesty, integrity, compassion and excellence. These principles equate with recognised qualities needed for transition and are associated with the three underpinning components of the SDT [20]. These components include Competence (the ability to complete tasks), Autonomy (the capacity to self-direct learning) and Relatedness (how well a person can connect with others).

Pupils were empowered to be responsible for planning, organising, executing and reviewing naturally emerging experiences beyond primary-level learning, even when things did not go to plan. For example, this included pupils moving from describing movement and applying basic problem-solving to selecting and appraising different components of physical education across activity areas. While these activities were novel for many pupils, they were not so far removed from every-day school life to make them seem irrelevant and non-transferable. Therefore, within the limits of safety, pupils perceived being directly responsible for shaping their learning outcomes and for the nature and direction of the activities undertaken. This included pupils working collaboratively and in negotiation with facilitators, whether building rafts, organising equipment for journeys or ensuring personal and collective safety. 

Further, to bridge to the secondary school curriculum, protocols for OA activities were infused with learning tasks from physical education, numeracy, literacy and information technology Although qualified OA practitioners delivered the technical elements of the programme, 10 undergraduate student volunteers facilitated all other collaborative activities such as a group-planned nature-based journey, housekeeping duties and meal preparation. Most importantly, these university students acted as an interface between school teachers and the children. Throughout, photographic images from previous days’ events, written reflections and pictures associated with a range of emotions were displayed; group presentations allowed pupils to present evidence of their learning through role play, songs, poetry, narratives and poster work. Themes emanating from this work included (i) dealing with fear of the unknown, (ii) inclusion, (iii) empathy for others, (iv) sharing, and (v) future desires to go to university. 

#### 2.3.2. Comparison Programmes

Intervention studies within education should establish realistic comparisons [56]. To provide suitable comparisons for this programme, two induction programmes aimed at supporting children in their transition were evaluated. The first comparison group experienced an active one-week induction programme delivered in school. Activities focussed on integrating pupils into their new environment, familiarising them with subject areas, and helping them to form friendships. A second comparison group involved children who attended a five-day commercial OA residential programme. This comprised team building challenges, land- and water-based pursuits. There was no attempt to align this programme to the school’s educational objectives or aspects of self-determination and general well-being. All activities were delivered by experienced qualified personnel. The characteristics of all programmes are featured in Table 1


### 2.4. Measures and Data Analyses 

#### 2.4.1. Quantitative 

Two validated age-appropriate self-report questionnaires were completed by children immediately before and on completion of their respective programme. The 14-item Warwick–Edinburgh Mental Well-being Scale (WEMWBS) [57] provides a single graduated score reflecting pupils’ positive thoughts and feelings. The 21-item Basic Psychological Needs Satisfaction in Life Scale (BPNS) [20] depicts pupils’ self-determination through three separate subscale scores of autonomy (freedom to express ideas), competence (ability to learn interesting new skills) and relatedness (amount of care received from others). Both the WEMWBS and the BPNS have established age-appropriate validity and reliability, possess positive links to increased psychosocial and academic functioning of pupils in schools [58,59] and relate closely to behaviours needed for successful transition. 

#### 2.4.2. Qualitative 

This approach enabled qualitative data to strengthen inferences contained in quantitative findings of the Tailored OA programme. Semi-structured interviews and informal discussion took place in groups of five/six children during and following the residential programme (immediately and four months later in school). Children were encouraged to express their thoughts and opinions through open questioning and discussion which allowed clarification and exploration of ideas [60]. Questions were guided by elements which underpin psychological well-being and components of self-determination (Table 2). Teachers took part in the open interviews to share their perceptions of the value of the intervention programme for the schoolchildren’s general well-being and their ability to respond to the expectations of the school. 

#### 2.4.3. Data Analyses 

Quantitative data analyses investigated the magnitude and direction of change to measures of psychological well-being and self-determination immediately following all three induction programmes. Similarities and differences within and between group (mean) scores for WEMWEBS and BPNS were identified using descriptive and parametric statistical analyses (percentage differences, independent t tests, effect sizes, one-way between-participants analysis of variance). Analyses were conducted using Statistics Package for the Social Sciences (SPSS) Version 24 [61]. In-programme and follow-up qualitative data analyses involved transcription and thematic analysis (a coding technique allowing information to be sorted into distinct frameworks which related to the research aims and objectives). 

## 3. Results

### 3.1. Quantitative Data

#### 3.1.1. Psychological Well-Being 

A significant difference was observed between the three programmes (Tailored OA, School Induction, Generic OA) on pre- and post-programme differences in psychological well-being F (30,69) = 1.97 < 0.05 (Figure 1). The Tailored OA programme achieved the greatest improvement in psychological well-being (‘medium’ effect size 0.43) compared to the school-based induction, which reported a reduction, and the generic programme, which registered a small improvement. 

#### 3.1.2. Self-Determination 

Figure 2 highlights the degree of differences in the mean BPNS subscale scores of autonomy, competence and relatedness for each programme. Beneficial increases in each subscale were evident for the Tailored OA residential programme. These subscale increases represented an overall effect size increase of 0.25 for self-determination. In contrast, the school-induction and generic OA programmes recorded decreases in Autonomy and Competence and smaller increases in Relatedness. 

The tailored OA programme recorded statistically significant differences in Autonomy (effect size increase of 0.25) compared to the school induction programme, t (78) = 3.04, p = 0.05, and the generic OA programme t (78) = 3.42, p = 0.05. The tailored OA programme also increased the Competence of school children (effect size 0.18.) There was a statistically significant difference in Relatedness between the Tailored OA programme (effect size increase of 0.33) and the generic OA programme t (78) = 2.84, p = 0.05. 

### 3.2. Qualitative Data

#### 3.2.1. Psychological Well-Being 

Qualitative responses of children attending the tailored OA programme confirmed findings from the WEMWBS measure. Children portrayed confidence in their abilities, contentment and appreciation for others during a programme which emphasised active immersion within nature.

*“I was proud of raft building,* *I kept falling off but learnt how to climb back on”**“I didn’t think I was a good leader,* *leading orienteering”**“I learned that you can show compassion without even realizing it!* *I know that I can show it my friends**now.* *I feel happier”**“I couldn’t believe I was outside so late.* *I love being outside”**“I enjoyed learning outside* *because I didn’t even think I was learning… it’s boring in the classroom.”*

#### 3.2.2. Self-Determination 

Children attending the tailored OA programme reflected upon their enhanced self- determination depicted through the three subscales of autonomy, competence and relatedness. 

In respect to the development of the subscale of Autonomy, children recognised opportunities for self-reliance through authentic challenges.

*”In map reading* *we found our way back from the river without help from the grown-ups!”**“I felt independent* *when we had to clean our rooms and make our own sandwiches”**“It didn't matter that* *we got wet when our raft collapsed, we just re-built it”*

Children considered their improvements in the subscale of Competence following successful negotiation of the tasks presented through their perseverance and effort.

*“In archery I* *came first but I didn’t think I would, and made me more confident in my ability”**“Having done* *the residential I think I can cope with this [secondary school] responsibility because I know**I am capable* *of it”**“My favourite* *thing was doing a presentation, this made me feel excited to do it again at school and be not**so* *scared”**“Sky-walking was* *really scary but I did it with help from my new friends”*

The subscale of Relatedness increased across all groups. However, the sharpest increase was reported within the tailored OA programme where the importance of collaborative effort and support for others was continuously reinforced. This equated to future challenges children would face in school. 

*“I found working together meant it were easier—If I was on my own,* *I wouldn’t have done it”*. *“I think that I have lots of people* *to talk to now and I can go to my teachers”**“We have more friends* *because we slept in the same room and did activities together so we helped and**supported all the* *time”*

#### 3.2.3. Teachers Perspectives on Tailored OA Programme 

Teachers’ perspectives helped to illuminate processes associated with learning within the tailored OA programme. The characteristics of the tailored OA programme placed emphasis on allowing the children the freedom to plan and explore, undertake supported risk-taking and review naturally emerging experiences.

*“There are no right or wrong answers, just a process, with multiple solutions.* *In working through these,**children are able develop creativity,* *collaborative learning and decision-making skills so early in coming**to* *school”**“If the trust and relationship is not there,* *the pupils do not have much confidence in the classroom (or in**you) which links into lower* *academic attainment”**“Usually we see pupils only twice* *a week for an hour and so it takes longer to form relationships...gaining**trust and understanding of how they* *learn may take till Christmas—this is a way to get them on track**before* *then”**“Unfamiliar activities act as a leveller,* *whereby some children who traditionally are more dominant in**school may be stretched out of their comfort zones outside,* *those quiet kids get a chance to shine”*

#### 3.2.4. Tailored OA Programme Sustainability of Impact

Four months following the programme, children and teachers were able to self-reflect on the importance of these behaviours which showed a degree of resonance. 

*“When I started this school,* *I was really shy but now my confidence has grown because I got to know**people better than if we didn’t* *go on residential”**“Yeah, I remember the time we had at the Outdoor Centre,* *and when I start to feel nervous, I remember how well I got on and how you have to try something even if it is scary”**“Practice makes perfect as I learn in archery that’s because I saw improvements in me,* *so I practice much**more now with* *other things”*

Behaviours regarded as important for transition were observed by teachers during school time. 

*“Pupils were* *drafting and re-drafting and they weren’t happy or content with it being**mediocre......sometimes it* *took six or seven attempts.......not by the teacher saying it isn’t good enough, it**was the students* *taking responsibility over their work and being proud of what they had done and*
*achieved”*
*“They don’t* *[pupils from the intervention group] seem to have the [academic] dip as much, they are more**confident.* *They ask for help much more and seem a lot happier around school—and attendance is better”*

## 4. Discussion 

This study investigated the efficacy of a tailored OA programme for facilitating benefits in children’s psychological well-being and self-determination during and following their transition into secondary school. Investigations suggested a tailored OA programme compared to a non-OA school-based induction programme and generic OA intervention achieved the strongest scale of change in psychological well-being and in all three SDT subscales of autonomy, competence and relatedness. Increases in the perceived ability of school-children to connect with others during transition were reported across all three induction programmes. Qualitative testimonies corroborated the quantitative findings of the tailored OA programme, highlighting personal experiences and processes underpinning these changes.

Although limited to a modest population sample, these findings shed light onto the benefits of developing focused strength-based functioning within an OA residential programme for school children. More importantly, the nature of this change suggests that interventions can be devised that, potentially, support an effective transition for children from inner-city areas who may not be able to access effective learning in green spaces. OA residential programme exposure which helps pupils to (i) feel proud and content (well-being) (ii) become independent (autonomy), (iii) be good at something (competence) and (iv) feel valued as a group member (relatedness) can produce a range of adaptive capabilities that help transition to secondary school. To discuss the implications of these major findings, each of the study's objectives are considered separately. 

### 4.1. The Psychological Well-Being of Schoolchildren across Programme Conditions 

Both OA programmes delivered short-term increases in the children's psychological well-being compared to induction practices undertaken in a school setting. Items contained on the WEMWBS scale include the extent to which children ‘feel optimistic’, are ‘interested in others and new things’, can ‘deal with problems’, and ‘feel loved’. In this regard, these findings confirm the value of the exposure of young people to novel, shared activities in a natural residential OA for providing immediate psychosocial benefits (‘social capital’, creativity, a sense of belonging) (e.g. [28,29,35,37]) that may transmit into school life and beyond [40]. Findings also suggest that natural settings advocating shared expectations, freedom of expression and promotion of teacher–pupil relationships may be preferable for embedding new pupils in transition than more uniform environments [50].

### 4.2. The Self-Determination of School-Children across Programmes 

Intense challenges in OA which emphasise the need for self-reliance may create real senses of capability through individual’s overcoming dissonance [21]. The tailored OA programme was foremost in enabling changes to more schoolchildren's self-determination. This aligns with findings from a recent similar study of children transiting into secondary school [50]. The predominance of change across all subscales reported by children exposed to this programme provides justification for deploying focused, collaborative approaches in OA for addressing schoolchildren’s transitional needs. Appropriately planned and executed programming involving schools and wider partners may help to inform teachers and create confidence in formulating OA. This practice may take the form of distinctive residential programming or curriculum-based outdoor learning which is delivered in and around school premises. 

### 4.3. Processes Associated with Learning within the Tailored OA Programme 

To prepare incoming schoolchildren for the reality of secondary education, schools have been encouraged to develop approaches for pupils to become more self-determined [16]. The SDT subscale of Relatedness increased across all groups. This could represent the overarching emphases placed upon social skills needed for transition within each of these programmes. Nevertheless, the characteristics of the tailored OA programme placed most emphasis on allowing the children the freedom to plan and explore, undertake supported risk-taking and review naturally emerging experiences. To consolidate learning, children were encouraged to move from describing outcomes and applying basic problem-solving (primary learning) to selecting, appraising and presenting an understanding of skills needed to achieve in school. Although these skills aligned with the school’s philosophy (i.e., honesty, integrity), they were more practically understood as making friends, knowing staff, asking for help and being responsible for oneself and others.

### 4.4. The Sustainability of the Tailored OA Programme

It is contested that exposure to OA experiences does not implicitly build positive characteristics in young people which transfer across contexts but provide situations whereby individuals experience novelty and/or feel compelled to conform [52]. Although the transferability of OA continues to be questioned, there continues to be a dearth of evidence which advocates the use of OA for the holistic development of young people (e.g. [26]). In the present study, personal attributes akin to personality traits were not targeted for change through enforced participation. Rather, self-directed habitual behaviours in children were introduced and encouraged throughout all aspects of programming which could be replicated in local green spaces and school settings. 

### 4.5. Limitations and Future Considerations

This study provides valuable insights into the comparability of programme effectiveness for enacting changes to well-being and self-determination of children during and beyond school transition. However, there are caveats to these findings. A limited number of pupils were recruited from similar schools, and therefore the findings do not generalise across the sector. Neither do the data reflect the on-going demands of a full academic cycle. Furthermore, the data can only confirm the programme content and follow-up measurement for the tailored OA programme. However, the measures of well-being and self-determination were responsive in determining differences in pupils’ functioning and could be used to evaluate further impacts of targeted OA interventions upon school children in transition.

Evidence from the current study suggests that empowering children to taking responsibility for their own comfort, safety and learning through tailored OA programming provides authentic consequences which may lead to improvements in their well-being and personal adaptability. The children’s self-determination was ensured by channelling workable ratios of pupils towards self-directed tasks, allowing them the freedom to succeed and fail in a neutral testing ground for challenges akin to those faced in secondary school. These findings provide encouragement for schools delivering active programmes aiming to smooth the transition of school children, particularly those targeted at more vulnerable groups. These experiences are optimised when teachers collaborate with experienced OA providers to shape programming to meet the specific aims of schools. 

## Figures and Tables

**Figure 1 sports-07-00134-f001:**
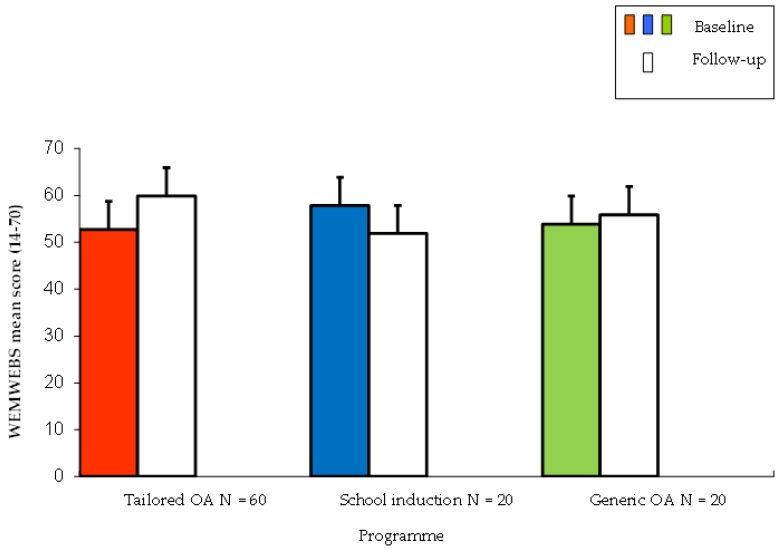
Baseline and follow-up Warwick–Edinburgh Mental Well-being Scale (WEMWBS) score by programme. One way Anova F(30,69) = 1.97 < 0.05 (Tailored OA v School induction and Generic OA) T bar lines on each block depicts the standard deviation of mean scores for each programme.

**Figure 2 sports-07-00134-f002:**
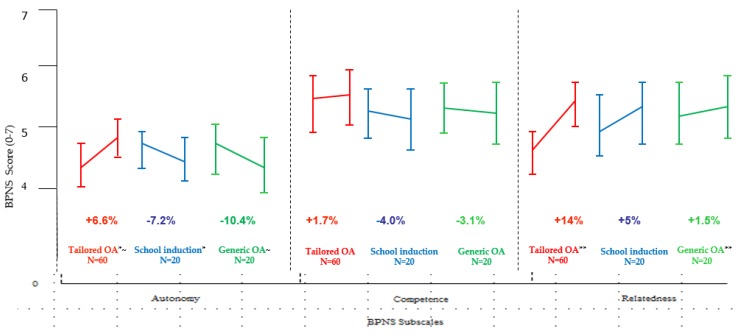
Baseline and follow-up mean difference scores on Basic Needs Satisfaction in Life Scale (BPNS) for Autonomy, Competence and Relatedness by programme. BPNS 0-7 (low-high) standardised scale for each subscale mean score; Independent t tests * Tailored OA v School induction t(78) = 3.04, p = 0.05; ~Tailored OA v Generic OA t(78) = 3.42, p = 0.05; ** Tailored OA v Generic OA t (78) = 2.84, p = 0.05; Lines depict standard deviation.

**Table 1 sports-07-00134-t001:** Programme Characteristics

Characteristics	Tailored OA	School Induction	Generic OA
Participants Sample size School location	Transiting pupils 60 Urban	Transiting pupils 20 Urban	Transiting pupils 20 Urban
Duration	3 days, 2 nights	5 days	5 days, 4 nights
Programme	Bespoke OA Residential programme	Non-residential school programme	OA Residential programme
Activities	A carousel of activities. Team-building challenges, land- and water-based. Strongly linked to the school’s core values. Emphasis on helping children to become self-determined. Pupil presentations	Class-based activities subject-specific lessons including ice-breakers, team-building challenges and practical activities	Team-building challenges, land- and water-based. No attempt to tailor programme to the school’s educational objectives or aspects of self-determination
Delivery/Staff	OA activities delivered by qualified instructors fully supported by teachers. University students fully immersed in all activities as mentors	Whole programme delivered by teachers	All activities delivered by qualified instructors. Teachers provided pastoral care and evening supervision

**Table 2 sports-07-00134-t002:** Sample of themed discussion questions

Can you talk about some of the new friends you have made this week ****	Can you explain how you feel about yourself generally, do you feel valued, and close to other people *	How competent do you think you’ll be, looking after yourself at school? How will you cope with the timetable and homework? ***	Can you explain how you feel in terms of your energy, and how cheerful, relaxed and positive you are feeling*
Can you tell me what you have learned this week (tell me a story) ***	Please indicate your levels of confidence and how interested you are in new things*	If you have a problem at school can you tell us who you would go to for help? **	Do you feel like you can take on the challenge of being at a new school? Please explain your answer***
How confident are you that you can cope with new subjects you are studying at school? **	How do you think you will relate to your teachers as a result of this induction programme? ****	Can you rate yourself on how clear your thinking is and your ability to solve problems *	How do you think you will you cope with the changes you face in your school schedule?**
What are your main sources of support (who can you talk to) at school, at home, outside? ****	How independent do you feel as a result of your experiences on this programme? ***	Can you talk about something from this week that has challenged you/made you very proud of yourself? **	Can you explain if and how being on residential has helped you to get along with people more effectively? ****

Key: Psychological well-being* & Self-Determination Theory (SDT) subscales of Autonomy** Competence*** & Relatedness****.

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
