# Peer review of "Purposeful Outdoor Learning Empowers Children to Deal with School Transitions"

_sports, 2019, doi:10.3390/sports7060134_

Reviewer 1 Report

This study is about an important topic. Overall, the authors did a good job of describing the research.

The main weakness with this paper is the writing quality. As I read this paper, I found  sentences that need improvement. The authors need to correct the problematic sentences before this paper can be accepted for publication. Below, I offer some examples of sentences that need improvement:

On page 1: 

Moving from a small school where lessons take place in a single room with one teacher, to navigating a complex timetable with multiple subjects, locations and teachers, provides a challenging proposition.

This is an awkward sentence.

On page 2:

Schools who report successful transitions focus on helping children to become more self-reliant,

develop new friendships, familiarising them with new routines and promoting curriculum interest

and continuity (14,15).

School that, not schools who. This is not the only problem with this sentence. The authors need to make sure the manuscript is free from problematic writing like this.

On page 2:

Additionally, a growing number of studies suggest just being in close proximity to ‘green’ spaces

significantly contribute to children’s improved mental wellbeing, reduced anxiety and behavioural

issues which are long-lasting

contributes, not contribute. The subject of the verb here is "being in close proximity to"

Page 13

Evidence from the current study suggests that empowering children to taking responsibility and

control for one’s own comfort, safety and learning through tailored OA programming provides

authentic consequences which may be linked to improvements in their well-being and personal

adaptability.

"One's" needs to be "their" because the authors are referring to children which is plural.

Page 13

This was ensured by channelling workable ratios of pupils towards self-directed tasks;

 allowing them the freedom to succeed and fail in a neutral testing ground for challenges akin to

those faced in secondary school. This provides encouragement for delivering active programmes

aiming to smooth the transition of school children, particularly those targeted at more vulnerable

pupils.

In these 2 sentences, it is not clear what "this" is referring to. This what? The authors need to be specific here.

Author Response

As requested, the writing quality of the paper has been improved, providing more informed, concise sentences.

For example,

Page1 awkward sentence now reads:

Challenges include moving from a small school where lessons take place in a single room with one teacher, to navigating a complex timetable with multiple subjects,locations and teachers.

Page 2:

Schools that report successful transitions help children to develop new friendships, become more self-reliant, and promote curriculum interest and continuity (14,15).

 Additionally, a growing number of studies suggest just being in close proximity to green spaces significantly contributes to improvements in children's mental health and well-being (37,38,39,40).

Page 13:

Evidence from the current study suggests that empowering children to taking responsibility for their own comfort, safety and learning through tailored OA  programming provides authentic consequences which maay lead to improvements in their well-being and personal adaptability.

The children's self-determination was ensured by chanelling workable ratios of pupils towards self-directed tasks; allowing them the freedom to succeed and fail in a neutral testing ground for challenges akin to those faced in secondary school.

These findings provide encouragement for schools delivering active programmes aiming to smooth the transition of school children, particularly those targeted at more vulnerable groups.

Reviewer 2 Report

Thank you for the interesting manuscript and effort you have put into this project. Find below a series of comments that will help to strengthen the manuscript.

The abstract would benefit from some simple statistical data at least showing significance as well as a list of the qualitative themes.

Line 115 – why not have a true control group (e.g. no OA in any form)? This should be mentioned in the limitations section

Line 131 – can more detail be added on the recruitment? How about specific ethnic and SES information?

Line 141 – 2 of 3 teachers should be 67%?

Could a detailed procedures section be added that walk the reader through the steps of the study?

It would also be helpful to split the measures section and out to have both a measures and analysis section. Also, please add effect size to the quantitative outcomes

Throughout the quantitative results section – please ensure complete statistical findings (t test results, above results, etc.) Also, as mentioned earlier, please add effect sizes to these results

I recommend moving the qualitative findings to the results section, highlight the themes and then add the data to support them.

The discussion would then focus more specifically on the relationship to previous findings and the practical implications of the results.

Author Response

Reponses to the reviewer are detailed below:

The abstract now benefits from simple statistical data showing significance and qualitative themes.

Line 115 – The school-based group is a non-OA true control group. This is made more expicit now throughout the paper 

Line 131 – Recruitment and SES information added 

Line 141 – 2 of 3 teachers is now 67%

A detailed procedures and study design section helps the reader understand the dynamics of the study

A separate measures and analysis section. Effect size calaculated and added to the quantitative outcomes

Complete statistical findings including effect sizes are detailed within the quantitative results section

Qualitative findings (themes and data) relocated to the results section. 

Discussion focuses on the relationship to previous findings and the practical implications of the results.